 eLIFE

# Experimentally guided models reveal replication principles that shape the mutation distribution of RNA viruses

Michael B Schulte[1,2†], Jeremy A Draghi[3†], Joshua B Plotkin[3], Raul Andino[2*]

[1]Tetrad Graduate Program, University of California, San Francisco, San Francisco, United States; [2]Department of Microbiology and Immunology, University of California, San Francisco, San Francisco, United States; [3]Department of Biology, University of Pennsylvania, Philadelphia, United States

**Abstract** Life history theory posits that the sequence and timing of events in an organism's lifespan are fine-tuned by evolution to maximize the production of viable offspring. In a virus, a life history strategy is largely manifested in its replication mode. Here, we develop a stochastic mathematical model to infer the replication mode shaping the structure and mutation distribution of a poliovirus population in an intact single infected cell. We measure production of RNA and poliovirus particles through the infection cycle, and use these data to infer the parameters of our model. We find that on average the viral progeny produced from each cell are approximately five generations removed from the infecting virus. Multiple generations within a single cell infection provide opportunities for significant accumulation of mutations per viral genome and for intracellular selection.

*For correspondence: raul.andino@ucsf.edu

†These authors contributed equally to this work

Competing interests: The authors declare that no competing interests exist.

## Introduction

RNA viruses are excellent models for evolution. They replicate quickly and have extremely high mutation rates, orders of magnitude greater than those of most DNA-based life forms (*Drake, 1993*). While this combination of traits creates the potential for rapid adaptation, it necessitates a life history strategy that balances the need for explosive, exponential growth with the requirement to maintain genomic integrity. The life history strategies of viruses are largely reflected by their mode of intracellular replication. Two classic replication modes have been described for single-stranded RNA viruses: the 'stamping machine' mode (*Stent, 1963*) and the 'geometric replication' mode (*Luria, 1951*). In the stamping machine mode (SM), templates made from the original infecting genomes are used for the production of all progeny genomes. In the geometric replication mode (GR), newly made progeny genomes are used to create further templates for additional rounds of replication within a single cellular infection cycle (*Figure 1*). Progeny produced from stamping machine replication are all a single generation away from the parental strand whereas progeny generated from geometric growth represent a distribution of generations from the parental strand, often resulting in a fractional mean number of generations (see *Figure 1*). The iterative nature of GR creates branched genealogies that allow for expansive exploration of sequence space and results in a mutation distribution that differs from the SM mode (*Luria, 1951*). Recent studies with population-genetic models (*Draghi et al., 2010*) and RNA enzyme populations (*Hayden et al., 2011*) have shown that differences in the distribution of mutants can significantly impact the adaptability of a population. Recent studies with poliovirus (PV) have also demonstrated that mutational differences within a population can have dramatic effects on pathogenicity (*Pfeiffer and Kirkegaard, 2005*; *Vignuzzi et al., 2006*) as well as fitness, virulence, and robustness (*Lauring et al., 2012*).

**eLife digest** Viruses with genetic information made up of molecules of RNA can multiply quickly, but not very accurately. This means that many errors, or mutations, occur when the RNA is copied to create new viruses. The advantage of this rapid, but mistake-filled, RNA replication process is that some of the mutations will be beneficial to the virus. This allows viruses to rapidly evolve, for example, to develop resistance against drugs.

The poliovirus is an RNA virus that can cause paralysis and death in humans. To prevent such infections, scientists have extensively studied the poliovirus and have developed effective vaccines against it that have eliminated the virus from all but a few countries. Because so much is known about the poliovirus and because it has a very simple structure, scientists continue to use the poliovirus as a model to study virus behavior.

One unknown aspect of the poliovirus' behavior is how it replicates after invading a cell. Are all of its RNA copies made from the original viral RNA that first infected the cell, in what is known as a 'stamping machine' model? Or do the new copies of the RNA also get copied themselves in a 'geometric replication mode' that increases the likelihood of mutations and enables the virus to evolve more rapidly?

Viral RNA molecules are copied by one of the virus's own proteins and so before the viral RNA can be replicated, it must first be translated to form viral proteins. When and where replication begins depends on the concentration of translated proteins around the RNA and so replication tends to begin in particular areas of the cell at different times. Schulte, Draghi et al. used mathematical modeling to create computer simulations of the number of polioviruses in a cell that take into account these time and space constraints. By including random elements in the model, the simulated behavior more accurately follows experimentally recorded data than previously used models.

The results of the model led Schulte, Draghi et al. to conclude that the poliovirus replicates by the 'geometric mode'; as new copies of the poliovirus RNA are made, each copy goes on to make more copies. This means that in a single infected cell there are multiple generations of RNA, and each generation may undergo distinct mutations that are passed on to the next set of RNA copies. In fact, Schulte, Draghi et al. found that the average virus released from an infected cell is the great-great-great-granddaughter of the original virus that infected the cell. With so many different generations of virus coexisting in a cell, there are a lot of opportunities for new genetic combinations to occur and for viruses to evolve new abilities.

Poliovirus' simple genomic architecture and medical importance have made it one of the most extensively studied viruses (*Racaniello, 2006*). However, despite decades of mechanistic studies and recent revelations of the importance of population structures, the replication mode and resulting mutation distribution have yet to be determined. PV therefore proves an excellent candidate for the rigorous construction of a computational model of virus replication to predict population structure and mutation distribution. A major feature of PV intracellular dynamics is that the genome participates in multiple reactions: translation, replication, and encapsidation. Its 7.5 kb genome contains a single open reading frame, which encodes 7 nonstructural proteins and 4 capsid proteins. Translation produces a single polyprotein, which is cleaved into individual functional viral proteins. Replication of the positive-sense genome by the virus-encoded RNA-dependent RNA polymerase produces a negative-sense strand, which is used as a template for further genome synthesis. Evidence suggests that the initial, infecting positive-sense genomes must be translated before they can replicate (*Novak and Kirkegaard, 1994*). The switch from translation to replication appears to be dependent on the concentration of a viral protein product, 3CD, which stimulates a transition from a linear, translating RNA to a noncovalently associated 'circular' RNA competent for replication (*Gamarnik and Andino, 1998*, *2000*; *Herold and Andino, 2001*). Encapsidation is thought to result from protein–protein associations of capsid pentamers with the RNA replication machinery and protein–RNA association of capsid pentamers with viral RNA (*Pfister et al., 1992*; *Nugent and Kirkegaard, 1995*; *Liu et al., 2010*). Actively replicating genomes are preferentially encapsidated and packaging is biased to exclude negative-sense strands, although the mechanism of this is not understood (*Nugent et al., 1999*). Although

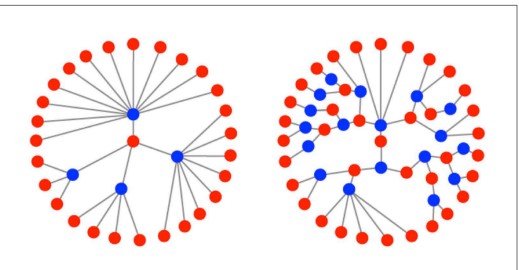

**Figure 1**. Illustrations of the genealogies of different replication modes. Red dots indicate positive-sense strands. Blue dots indicate negative-sense templates. Stamping machine (SM) progeny are one generation from the initial infecting genome (left). In an example of geometric replication (GR), progeny are an average of 2.33 generations from the initial infecting genome (right).

multiple ribosomes can translate a genome at the same time and multiple viral polymerases can replicate a genome at the same time, the processes are mutually exclusive (*Gamarnik and Andino, 1998*). Similarly, neither translation nor replication can occur after a genome is packaged into a virion.

Several studies have demonstrated that PV genomes are often localized to the cytosolic surfaces of the endoplasmic reticulum, Golgi bodies, lysosomes, or vesicles derived from these (*Schlegel et al., 1996*; *Bolten et al., 1998*; *Cui et al., 2005*; *Egger and Bienz, 2005*; *den Boon et al., 2010*). Replication complexes are thought to form on these membranes in *cis*, resulting in a close association of translation products and positive-sense genomes (*Novak and Kirkegaard, 1994*; *Egger et al., 2000*). Compartmentalization of replication complexes likely accounts for the observation that many functions of nonstructural proteins cannot be complemented in *trans* (*Novak and Kirkegaard, 1994*; *Ansardi et al., 1996*). Only capsid proteins, 3CD, and 3D have been demonstrated to *trans*-complement (*Novak and Kirkegaard, 1994*; *Nugent et al., 1999*; *Oh et al., 2009*). Taken together, these studies suggest that the essential transitions—from translation to replication, and from replication to encapsidation—are largely localized and influenced by the dynamics of the molecules in each compartment.

In recent years, modeling approaches have begun to examine the trade-offs that come with having a genome that is a template for both replication and translation (*Krakauer and Komarova, 2003*; *Regoes et al., 2005*; *Sardanyés et al., 2009*; *Thébaud et al., 2010*; *Martinez et al., 2011*). These studies have raised mechanistic and evolutionary questions about the life cycle of single-stranded, positive-sense RNA viruses, but most have not produced models that can be directly compared to data. Several previous models are deterministic in nature (*Krakauer and Komarova, 2003*; *Regoes et al., 2005*; *Martinez et al., 2011*) and assume a well-mixed, spatially uniform cellular environment (*Krakauer and Komarova, 2003*; *Regoes et al., 2005*; *Sardanyés et al., 2009*; *Thébaud et al., 2010*; *Martinez et al., 2011*). Experimental evidence suggests that each of these assumptions is problematic and do not reflect the biological constraints and properties of viral replication. The small numbers of the critical molecules that initiate an infection suggest that a stochastic model would more accurately describe early reactions and could make distinct predictions from previous deterministic approaches (*Srivastavawz et al., 2002*). Often infections begin with relatively few virions that release their genomes into the cell and continue with the translation of these few initial genomes. Random variation in the switch from translation to replication is amplified by the subsequent exponential phase of the infection, and this amplification is likely to bias the mean dynamics of a set of infections. Indeed, recent single-cell studies demonstrated the significant impact of stochastic effects on poliovirus infections (*Schulte and Andino, 2014*).

Here, we have developed a stochastic simulation model in which we compartmentalize reactions in an effort to accurately describe intracellular dynamics in both space and time. Additionally, rather than fixing each parameter on an estimated value, an approach used by previous models, we use an Approximate Bayesian Computation approach to fit our parameters from temporal quantitative data. We find that by combining stochasticity and spatial structure, our model reflects and describes the population dynamics and structure of the viral population during an infection cycle more accurately than previous models.

Fitting our model to RNA abundances over time, we find that poliovirus follows the geometric replication mode: multiple iterative generations of genomic replication produce progeny virus. Posterior parameter fits indicate that progeny of a single cellular infection are approximately five generations away from the initial, infecting genomes. This replication mode produces populations with expansive, branched genealogies, creating the dramatic potential for the exploration of sequence space, as well as creating the potential for intracellular selection among related mutant genomes.

## Results

### Inference of replication parameters

We used temporal, quantitative RT-PCR data of both positive-sense genomes and negative-sense strands to estimate the free parameters in our model. The role of each parameter in poliovirus replication and in the mathematics of our model are diagrammed in *Figure 2* and described in detail in the 'Materials and methods'. We chose to use measurements of positive- and negative-sense RNA at multiple time points for three multiplicities of infection (1, 10, and 100), as well as measurements of virion numbers at multiple times for MOI 10; this amounted to 27 measured means, with three data points for each mean. Strand-specific qRT-PCR was performed to quantify positive-sense and negative-sense poliovirus RNA against in vitro transcribed standard RNAs of each sense (*Burrill et al., 2013*). Along with cell counts, this allowed for temporal measurements of the average positive-sense and negative-sense poliovirus RNA copies per cell. Negative-sense RNA was not detectable until 2 hr post infection for MOIs 10 and 100 and 3 hr post infection for MOI 1. Positive-sense RNA was clearly quantifiable for all time points at the MOI 10 and 100 but did not rise above background levels until 3 hr post infection for MOI 1. Using a newly developed virion immunoprecipitation assay (*Burrill et al., 2013*), we observed de novo virion assembly between 2 hr and 3 hr post infection. Along with total positive-sense RNA measurements from this time course, we obtained a percentage of genomes encapsidated in quadruplicate at 3 hr, 4 hr, and 5 hr post infection. *Figure 3* illustrates this data alongside projections from inferred parameters from the second round of SMC (see *Figure 3—source data 1*).

The relatively high number of data dimensions, combined with the computationally intensive and highly stochastic nature of our simulations, made a traditional maximum likelihood approach impractical. Instead, we turned to Approximate Bayesian Computation, using as our summary statistic the sum of the squared deviations of the average simulated RNA concentrations (and average fraction of virions for MOI 10) from their corresponding empirical means. This algorithm produces progressively more accurate estimates of each parameter over several rounds; *Figure 4—figure supplement 1* illustrates that, for most parameters, round one restricts the credible range of each parameter in

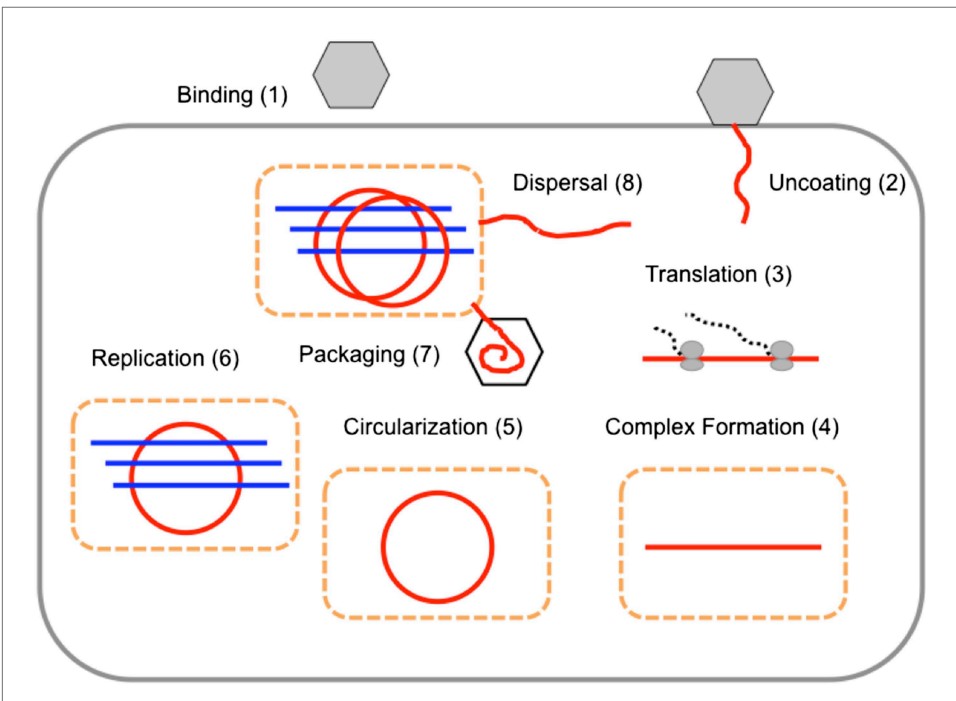

**Figure 2**. The replication cycle of poliovirus as represented in our model. Numbered steps correspond to sections and equations in the 'Materials and methods'.

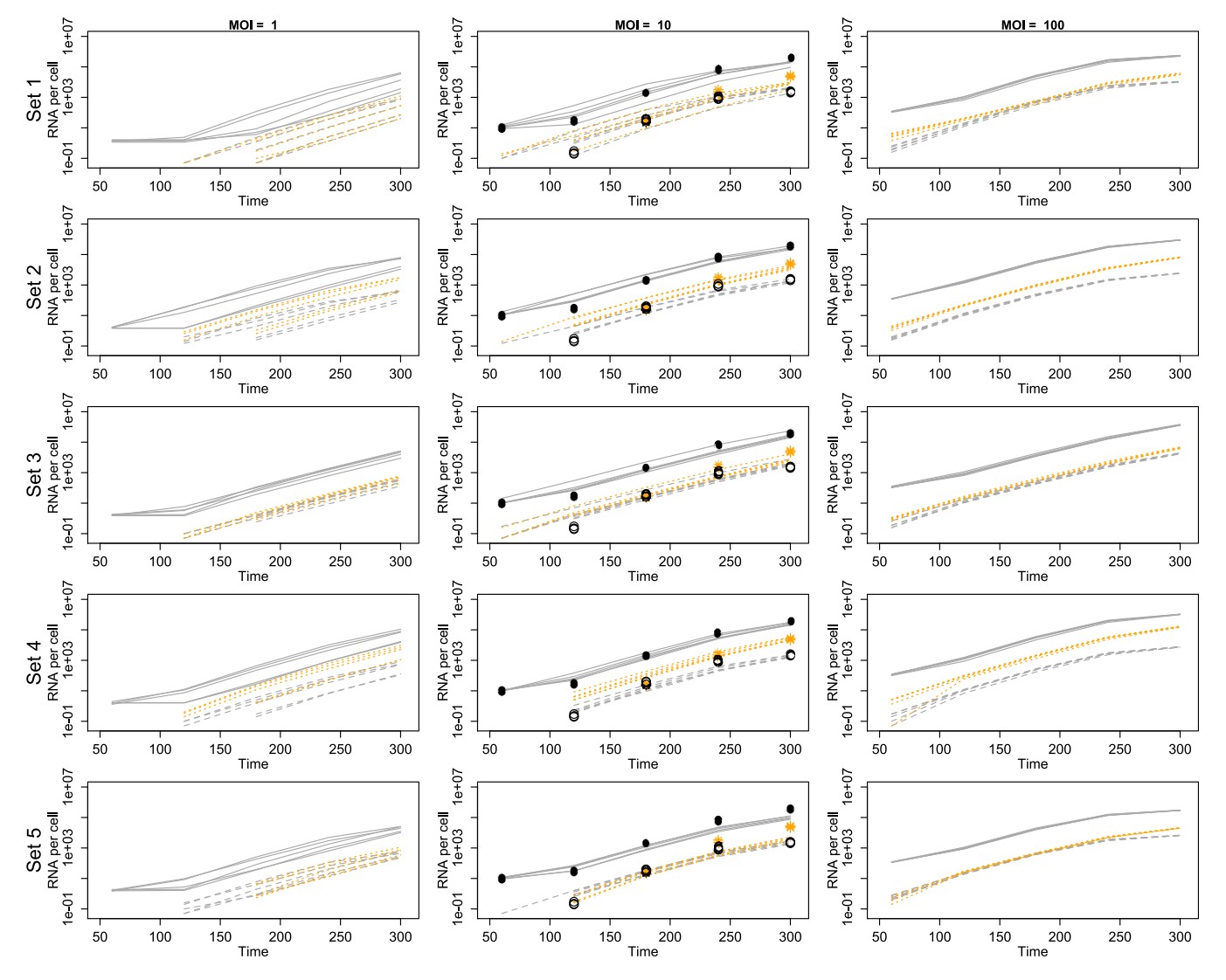

**Figure 3**. Projected mean abundances of positive-sense RNA (solid line simulations vs filled circle experimental measurements), negative-sense RNA (dashed line simulations vs hollow circle experimental measurements) and virions (orange dotted line simulations vs star experimental measurements; measured only for MOI = 10). Each row represents a different example parameter set (see 'Results'); each line is the mean of 20 individual cell simulations, and the means of five sets of 20 replicate simulations are plotted in each panel. Parameter values are given in **Figure 3—source data 1**.

The following source data and figure supplement are available for figure 3:

**Source data 1**. 'Best' parameter set used in **Figure 3—figure supplement 1**, and **Figure 4—figure supplement 4**.

**Figure supplement 1**. Distribution of virions in 10,000 replicates for simulations with (points) and without (line) a deterministic threshold for waiting times (see 'Materials and methods').

comparison to the flat prior and round two leads to further focusing. The data appear to be uninformative for at least one parameter, $c_{pack}$; a second parameter, $com_{max}$, appears to be poorly constrained by the comparison to MOI = 10 in round one, but somewhat constrained by the broader measurement against all three MOIs in round two. Round two also appears to significantly move the mode of two other parameters, $c_{com}$ and $c_{3A}$.

**Figure 4—figure supplement 1** indicates that ABC inference informed the values of nine of our ten parameters, but these marginal parameter distributions alone do not capture correlations between parameter values. **Figure 4—figure supplement 2** shows evidence of significant correlations, and

*Figure 4—figure supplement 3* shows that parameter sets drawn from the marginal distributions in *Figure 4—figure supplement 1* (i.e., uncorrelated parameter values) do a poor job of matching the data. While not unexpected, these significant correlations require that we work directly with the sampled parameter sets arising from our inference process, which is the approach we take below.

Each parameter in the posterior is supported over a significant range of possibilities. This remaining uncertainty reflects two factors: the data may be insufficient to determine each parameter, and the inference process may not have fully exploited the inferential power of the data. We took several approaches to quantify the sufficiency of the data and the effectiveness of the inference process. First, we measured the mean error of parameter sets when compared to the data for each multiplicity of infection independently; we asked if performance at one MOI predicted performance at the other two. If so, the dimensionality of our data would be effectively lower than we had initially assumed. Surprisingly, pairwise correlations between mean error at one MOI and another were very weak: Spearman's rho is 0.031 for MOIs 1 and 10, −0.092 for MOIs 1 and 100, and 0.129 for MOIs 10 and 100. This suggests that measurements at each MOI are contributing distinct information to our inference process.

Second, we determined the sensitivity of our measure of model 'fit' to variation in each of the parameters. This analysis, described in detail in the 'Materials and methods', showed that the data significantly informed the values of eight of ten of the parameters (*Figure 4—figure supplement 4*). We also performed a separate validation analysis which attempted to infer the replication phenotype, $\overline{g}$, from mock data simulated from parameter sets drawn from our prior distribution. As described more fully in the 'Materials and methods', this exercise confirms that the data and method are adequate to infer the trait of interest, albeit with some degree of inaccuracy (*Figure 4—figure supplement 5*).

Finally, we examine the fit between the data and the mean dynamics of inferred parameter sets. *Figure 3* shows that the inferred parameter sets generally capture the information in the RNA and virion data, although some parameter sets deviate consistently from the data for some values. Variability among replicate sets of twenty single-cell simulations is substantial, correlated across a time series, and greatest for the smallest MOI. Further inferential effort could improve either the accuracy of the mean predicted dynamics or the precision of replicate simulation dynamics, though *Figure 3* suggests that such improvements could only be modest. This variability is expected due to the stochastic nature of the simulations, and it may better reflect the biological noise of the infection (*Schulte and Andino, 2014*).

## Predicted replication dynamics

*Figure 4* shows the inferred posterior distribution of $\overline{g}$, the mean number of generations for a packaged virion based on two rounds of inference with measured RNA and virion abundances. This distribution is plotted for MOI = 10; the predicted values at MOI = 1 and MOI = 100 are very similar and highly correlated (weighted means: MOI = 1, 4.96; MOI = 10, 5.06; MOI = 100, 4.85; Spearman's rho (unweighted): MOI 1 and 10, 0.92; MOI 1 and 100, 0.85; MOI 10 and 100, 0.96). While this distribution does show substantial variance, it is strongly inconsistent with a 'stamping machine' mode of replication, which would have a $\overline{g}$ near one.

To explore the robustness of this inference, we compared the predicted dynamics of the model to an additional type of data: the fraction of positive-sense RNA molecules translating at each time point. We fractionated infected cell lysates and quantified positive-sense RNAs in monosome and polysome fractions relative to total positive-sense RNA copies. These data render a percentage of genomes associated with translation machinery and provide an additional set of data to evaluate the parameter sets produced by SMC. When measured at an MOI of 10 at 1, 2, 3, 4, and 5 hr post infection, the majority of positive-sense RNAs appeared to be associating with translation machinery, consistently averaging near 85%. Many of the inferred parameter sets are consistent with the measured values but a substantial fraction is clearly inconsistent (*Figure 4—figure supplement 6*). The summed squared error of the translating fractions is also correlated with $\overline{g}$ (*Figure 4—figure supplement 7*). To estimate how these new data inform our prediction of $\overline{g}$, we calculated a weighting factor based on the relative rank of the summed squared error of translating fractions, such that the parameter set with the best fit was assigned a weight of 1, the next a weight of 1134/1135, etc. Reweighting the distribution of $\overline{g}$ by this additional factor produced the distribution shown in *Figure 4*; the mean $\overline{g}$ shifts from 5.06 to 4.78.

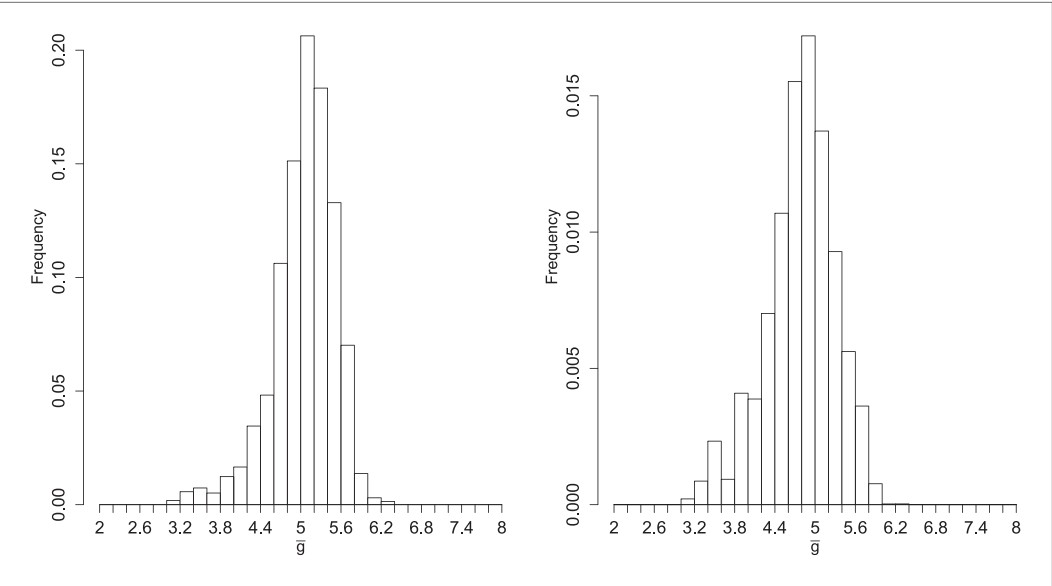

**Figure 4**. Left: posterior distribution of the mean number of generations of replication ($\bar{g}$). Right: distribution reweighted by the fit of predicted fractions of translating positive-sense RNA to empirical measurements.

The following figure supplements are available for figure 4:

**Figure supplement 1**. Prior and posterior distributions after each of two rounds of inference by Approximate Bayesian Computation.

**Figure supplement 2**. Correlations between parameters in the round two posterior.

**Figure supplement 3**. Log of total error for inferred, weighted parameter sets in round two (solid) vs 1000 sets assembled from parameter values drawn independently from the weighted posterior (dotted).

**Figure supplement 4**. Goodness-of-fit (1/[1 + mean error]) of highly replicated simulations for MOI = 10 and the 'best' inferred parameter set.

**Figure supplement 5**. Inference results from three validation experiments.

**Figure supplement 6**. Histograms of the projected fraction of positive-sense RNA undergoing translation for the mean simulated dynamics of each parameter set, compared to empirical measurements (orange dots).

**Figure supplement 7**. Summed squared error (SSE) of fraction of translating positive-sense RNA for all 1135 parameter sets, plotted against $\bar{g}$ at MOI = 10.

## Predicting the distribution of mutations

We simulated mutation and selection during infections to understand how replication dynamics shape the distribution of mutation frequencies among virions. To illustrate how mutant frequencies depended on $\bar{g}$, we chose two parameter sets with values of $\bar{g}$ at the low and high end of the range supported by the posteriors in *Figure 4* and included the 'best' parameter set as a representative of the more common values of $\bar{g}$. Mutation frequencies for these parameter sets ('best', 'low', and 'high'—see *Figure 3—source data 1*) are plotted in *Figure 5A* for a range of mutants that have a diminished rate of replication relative to the wild type. We chose to model this particular type of deficiency because we expected that replication deficits would directly affect the growth and packaging of the mutants. We observed that deficits in a different trait, the rate of complex formation, were effectively invisible to intracellular selection (the frequency of a mutation with an 80% reduction in complex formation was estimated to be reduced by 0.6–4.6% compared to a neutral mutation in the 'best' parameter set); we

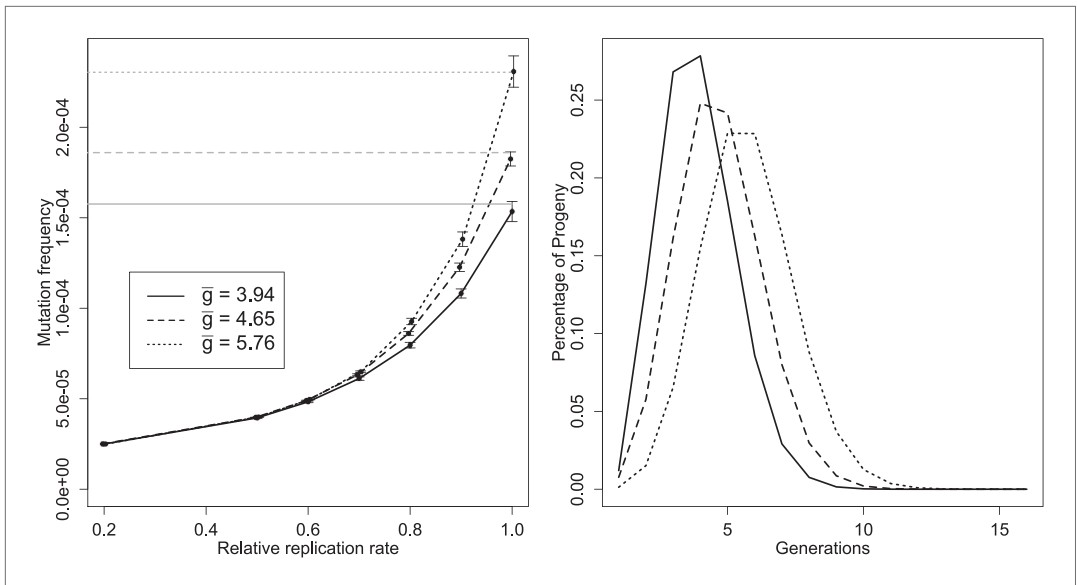

**Figure 5**. Left: mean mutation frequencies for three parameter sets ('low', $\bar{g}$ = 3.94; 'best', $\bar{g}$ = 4.65; 'high', $\bar{g}$ = 5.76). Mutation rate is 2 × 10$^{-5}$ per replication event; 'relative replication rate' reflects the reduced probability of a mutant template to replicate, relative to an unmutated strand. Grey lines indicate the expected mean for each parameter set with no selection (deficit of zero); the black line shows the mutation rate in one replication step, and therefore the expected frequency when mutants cannot replicate. Bars indicate 95% confidence intervals. Right: distributions of g of progeny from single cell infections for three parameter sets ('low', $\bar{g}$ = 3.94; 'best', $\bar{g}$ = 4.65; 'high', $\bar{g}$ = 5.76).

expect that mutations in traits like the rate of translation would also be complemented by the wild-type phenotype and so would not experience significant selection during the infection in which they arose.

Several distinct features of mutation in this model are evident from *Figure 5A*. Mutation frequency does not decrease linearly as intracellular selection approaches its maximal value; the curve results from the fact that mutant genomes that are immediately packaged are not subject to selection, while the contribution of rare, early mutants to average mutation frequency may be reduced by multiple rounds of intracellular selection. Knowing $\bar{g}$ and the mutation rate allows us to directly calculate the fate of neutral or very unfit mutations, but estimating the frequency of mutations of intermediate fitness requires additional simulations using our model. A third feature is the sizable confidence intervals relative to the number of infections sampled (10 million for each point). This high variability reflects the large contribution of very rare mutations that arise early in an infection and can contribute 1000s of mutant virions, especially when selection is weak.

The effect of these rare, early mutations in the overall mutant distribution can be seen as a departure from a Poisson process. To remove a potential confounding variation in burst size, we compare the distribution of mutations from infections within 10% of the median burst size and calculate a Poisson expectation for a median-sized burst with the same expected frequency. For the 'best' parameter set, median infections produced many more bursts with no copies of a given mutation (79.4% vs 22.5% for the Poisson), but also many more bursts with five or more copies of the mutant (8% vs 1.82% for the Poisson; $n$ = 51,365).

The distribution of the number of generations between progeny virions and initial infecting genomes is displayed for three parameter sets ('best', 'low', and 'high'—see *Figure 3—source data 1*) in *Figure 5B*. Only a very small percentage of progeny are produced via a single genomic replication cycle. Although all three parameter sets have means close to five generations, the distributions show a portion of the progeny virions representing up to 10 generations between the infecting genotypes and packaged virions within a single cellular infection.

## Discussion

The intracellular replication mode of a virus strongly influences the frequency and distribution of mutations among progeny, which shape the long-term behavior of an infecting population (*Vignuzzi et al., 2006*; *Lauring et al., 2012*). Due to the complex nature of intracellular dynamics, assessing the mode of replication of viruses is a difficult task (but see *Chao et al., 2002*). Here, we built on decades of mechanistic studies and recent modeling efforts to construct a stochastic computational model coupled with new Bayesian inference methods. We combined these mathematical and computational techniques with accurate temporal data to produce a detailed picture of viral infection. We found that positive- and negative-sense RNA measurements made across multiple MOIs, along with quantitative data on virion packaging, are sufficient to infer that poliovirus replication occurs in several layers of intermediate replication, in contrast to the oft-assumed 'stamping machine' model. The implications of the inferred geometric replication mode are as follows: (1) error rates per-replication are considerably lower than measured rates from full-replication-cycle in vivo studies, (2) for a given viral polymerase error rate, mutation will progressively accumulate in both genome and anti-genome RNAs, which should result in a more accentuated departure from the master sequence, allowing a better exploration of the available sequence space during a single infection cycle, and (3) there exists a significant potential for intracellular selection and competition among related genomes, even in infections initiated by only a single genome.

Accurate estimates of viral mutation rates are essential for studying viral evolution and have crucial practical applications in drug and vaccine design. While estimates of mutation rates exist for nearly two dozen viruses, estimates of replication modes exist for only a few (*Sanjuan et al., 2010*). Calculating per-replication event mutation rates from observed mutant frequencies is not possible, or even meaningful, without knowledge of the replication mode. Thus, estimates of poliovirus per-replication event mutation rates can vary over 10-fold depending on the assumed replication mode (*Drake, 1993*; *Sanjuan et al., 2010*). By inferring the mode of replication, we have been able to link estimates of per-replication event mutation rates to published mutant frequencies. The most extensive poliovirus mutant frequency data set estimated an average mutant frequency of $2 \times 10^{-4}$ (*Acevedo et al., 2014*). Using our inferred value of approximately five intracellular generations, we calculate a per-replication event mutation rate of $2 \times 10^{-4}/5 \times 2 = 2 \times 10^{-5}$, which is in agreement with the average estimates of poliovirus mutation rates calculated in vivo from lethal mutation frequencies (*Acevedo et al., 2014*). Rates of specific types of mutations, such as transversions and transitions, could each be inferred from their mutation frequencies by the same approach. Our inference of five intracellular generations is also in line with previous inferences of replication mode using the Luria-Delbruck fluctuation test null-class method (*Sanjuan et al., 2010*). However, our results highlight some limitations for inferring mutation rates from frequencies: intracellular selection may strongly affect mutation frequencies, and the strong stochastic nature of virus replication appears to deeply modulate minor allele distribution, which in turn will result in imprecise estimates of the expected frequency. In particular, assuming that mutation frequency can be modeled as a Poisson process will lead to inappropriate confidence in measured frequencies. As a consequence, multiple empirical mutation frequencies measurements will be required to obtain a more precise determination of true mutation frequencies.

The branched genealogy inferred in our study implies the potential for significant amounts of intracellular complementation, selection, and competition between mutant genomes, even in infections initiated by a single genome (*Novak and Kirkegaard, 1994*; *Turner and Chao, 1999*; *Vignuzzi et al., 2006*). *Figure 5A* demonstrates the extent to which the frequency of a mutation can be skewed by negative selection during the course of an infection. On the other hand, a mutational event that occurred early in replication and conveyed an intracellular replication advantage could potentially give rise to hundreds or thousands of descendant virions in a single generation. If the mutation distribution data in *Figure 5B* were displayed as a tree (as in *Figure 1*), it would contain over 7000 terminal nodes, too many to resolve in a figure. Hence, the apparent potential for mutant interactions is vast. These results suggest that the evolutionary fate of mutations may depend strongly on their intracellular competitive ability, even when multiplicities of infection are low. Additionally, studies that rely on bottlenecks to reduce selection in viral mutation studies (e.g., *de la Peña et al., 2000*) may be allowing more selection than anticipated. Future population dynamics studies should consider the implications of the intracellular expansion of mutant phenotypes.

Virus infections are normally depicted as deterministic processes that follow a stereotypical path from infection to progeny production and death of the infected cell. However, experimental data show that some infected cells produce few progeny while others produce large populations of progeny (*Schulte and Andino, 2014*). These observations support the notion that stochasticity is an important factor shaping the outcome of infection. By combining accurate experimental measurements with a stochastic model of viral replication, we have obtained a realistic description of how the molecular events driving the life cycle of the virus govern the outcome of infection in each cell.

A significant benefit of computational modeling is that the information learned in the empirical process of the development of a model can yield important insights in the biologic processes under study. For example, our initial attempts to fit temporal strand measurement data were unable to match the sharp transition to exponential growth seen in the data. Only after removing the requirement for positive-sense genomes to be translated before becoming replication-competent was our model flexible enough to rapidly create templates for exponential replication. While *Novak and Kirkegaard (1994)* demonstrate a requirement of the initial, infecting genomes to be translated before replication can occur, their data did not implicate that all genomes produced at any time during infection must be translated before replicating. Our study suggests that newly synthesized positive-sense genomes may or may not disperse to nucleate new replication complexes within a single cellular infection, allowing us to model intracellular dynamics in a novel way by permitting a portion of newly made positive-sense strands to immediately act as templates for replication without the requirement of translation.

Our model succeeds in describing many experimentally observed features of viral replication and is an excellent staging point for future and more accurate models of viral replication and evolution. With the realistic benefits of stochasticity, compartmentalized reactions, and parameters inferred from quantitative, temporal data, it acts as a baseline intracellular viral replication algorithm. More quantitative data, including data on the formation and number of replication compartments, would further inform the model. Potential additions of intracellular selection, complementation, and recombination parameters would allow population evolution studies to explore intracellular dynamics with more precision than previous approaches. The ultimate goal is to generate a comprehensive model incorporating mechanistic replication dynamics learned from virology with selection and complementation dynamics learned from population genetics. This tool could be very powerful for informing future therapeutic and preventative strategies.

# Materials and methods

## Experimental procedures

### Cells and viruses
HeLaS3 cells (ATCC CCL-2.2) were maintained in 50% DMEM/50% F-12 medium supplemented with 10% newborn calf serum, 100 U/ml penicillin, 100 U/ml streptomycin, and 2 mM glutamine (Invitrogen). Poliovirus Mahoney type I genomic RNA was generated from in vitro transcription of prib(+)XpAlong. To generate virus, 20 µg of RNA was electroporated into $4 \times 10^6$ HeLaS3 cells in a 4-mm cuvette with the following pulse: 300 V, 24 Ω, 1000 µF. The resulting virus was passaged at high multiplicity of infection (MOI ~1–10) three times then subjected to ultracentrifugation through a 30% sucrose cushion.

### Infections
Four wells of HeLaS3 cells in 12-well plates were washed, trypsinized, and counted twice each on a hemocytometer then averaged to determine cell count. To synchronize infections, plates were placed on ice, cells were washed with cold serum-free media and infected at MOIs 1, 10, and 100. Plates were incubated at 4°C for 30 min with rocking every 10 min to adhere virus. After removal of the inoculum, cells were washed 2× with warm serum-free media. Cells were then incubated at 37°C in 2% serum media until harvest. To harvest, plates were frozen at −70°C.

### RNA extraction, reverse transcription (RT), and qPCR
Plates were thawed on ice and refrozen at −70°C 3×. RNA was extracted via the PureLink RNA Micro Kit (Life Technologies) according to the manufacturer's instructions. cDNA was synthesized from total RNA using SuperScript III Reverse Transcriptase (Life Technologies) and 125 nM strand-specific RT primer (+strand_RT: 5′-GGCCGTCATGGTGGCGAATAATGTGATGGATCCGGGGGGTAGCG-3′;

-strand_RT: 5'-GGCCGTCATGGTGGCGAATAACATGGCAGCCCCGGAACAGG-3') in a 5-µl reaction. Separate RT reactions for positive and negative-strand RNAs were performed for each sample. RT products were treated with 0.5 units of Exonuclease I (Fermentas) to remove excess RT primer prior to qPCR. Strand-specific qPCR was based on a published protocol (*Burrill et al., 2013*). cDNAs were analyzed by qPCR using 2× SYBR FAST Master Mix (Kapa Biosystems), 200 nM strand-specific qPCR primer (+strand_For: 5'-CATGGCAGCCCCGGAACAGG-3'; -strand_Rev: 5'-TGTGAT GGATCCGGGGGTAGCG-3'), and 200 nM Tag primer (5'-GGCCGTCATGGTGGCGAATAA-3') in a 10-µl reaction. A 10× dilution series of in vitro transcribed positive- and negative-strand RNA standards was run alongside experimental samples and used to construct a standard curve.

## Virion immunoprecipitation
Lysates from MOI 10 infections were homogenized with a final concentration of 0.06% NP-40. Immunoprecipitation was performed using Protein A-coated Dynabeads and anti-poliovirus antibody according to a published protocol (*Burrill et al., 2013*).

## Sucrose gradients
HeLaS3 cells were infected for 1, 2, 3, 4, and 5 hr at an MOI of 10 in 15-cm dishes then simultaneously treated with 100 µg/ml cycloheximide (CHX) for 2 min at 37°C. Cells were washed with PBS+CHX and lysed with 0.5% NP-40 lysis buffer containing CHX on ice for 20 min. Cell debris was pelleted in a table-top centrifuge at 2000 rpm for 10 min at 4°C, then nuclei were pelleted at 9000 rpm for 10 min at 4°C. Cell lysates were loaded on a 10–50% sucrose gradient containing CHX and ultracentrifuged at 35,000 rpm for 3 hr. Fractions were collected on a Biocomp Gradient Station with a BioRad Econo UV Monitor. Fractions were pooled based on the spectrophotographic trace into two fractions (ribonucleoprotein and monosome/polysome fractions), RNA was extracted and subjected to qRT-PCR.

## Modeling replication
### Outline of stochastic simulation
We developed a stochastic simulation model that tracks discrete abundances of poliovirus molecular species within a cell and simulates individual reactions. This model is based on the Gillespie algorithm (*Gillespie, 1976*). In this model, stochastic events, such as the production, decay, or transformation of molecular species, are represented by reaction rates. Each rate can depend on $x$, the vector of abundances of all species in the system. Given an initial state $x_0$ at $t = 0$, the algorithm proceeds for a set duration ($t_{max}$) as follows:

 a. Sum the rates of all reactions 1..$n$ in the system; $r_{total} = \sum_{i=1}^{n} r_i(x)$.

 b. Draw the time until the next event, $\Delta t$, from an exponential distribution with a mean of $r_{total}$.
 c. Advance to time $t + \Delta t$.
 d. Choose which event occurred by drawing from a multinomial with probabilities $r_i(x)/r_{total}$.
 e. Change $x$ to reflect the chosen event.
 f. If $t < t_{max}$, return to step (a).

Similar to *Hensel et al. (2009)*, we have modified the basic Gillespie algorithm to balance accuracy and speed. When the $r_{total}$ is below a threshold (1000 events/min), we draw exponential times as described above; when it is above this threshold, we use the inverse of the rate—the expected time—as our interval between reactions. *Figure 3—figure supplemental 1* shows that this approximation delivers accurate results for the best (lowest error) inferred parameter set. This procedure allows us to efficiently generate stochastic realizations of replication, translation, and other reactions unfolding in a single infected cell, based on a system of equations that describes each essential reaction in the poliovirus life cycle. Results from many replicate simulations are then averaged to predict the dynamics across a population of infected cells.

*Figure 2* depicts the events in poliovirus replications captured quantitatively by our model, each of which is described in detail below.

### Binding (step 1)
We assume that the number of virions that bind to, and subsequently infect, a cell is Poisson distributed with a mean equal to the multiplicity of infection (MOI). This formulation assumes that bound

virions do not interfere with the binding of additional virions during the period of infection. We denote these coated positive-sense RNA genomes at $RNA^+_{initial}$; their distribution is therefore:

$$RNA^+_{initial} \sim Poisson(MOI).$$ (1)

## Uncoating (step 2)

A quantitative description of uncoating was derived from the data presented in *Brandenberg et al. (2007)*; based on these data, we choose the two-parameter gamma distribution to model stochasticity in this process. To account for differences in experimental protocols, we excluded the t = 0 measurement from Brandenberg et al.'s data and, taking the t = 8 min measurement as the starting point, fit the gamma distribution to the average cumulative measurements. Using the optim() function in R, we obtained an estimate of 0.678 for the shape parameter, and 0.02 for the rate parameter ($n$ = 28, $R^2 \approx 0.92$). Each of the $RNA^+_{initial}$ molecules transitions to a translationally competent, linear-form positive-sense RNA, $RNA^+_{lin}$, after a waiting time, $t_{uncoat}$, drawn from *Equation 2*.

$$t_{uncoat} \sim Gamma(0.02, 0.678).$$ (2)

## Translation (step 3)

Translation is the first role of positive-sense genomes in a cell, and it continues as the primary role throughout the infection. Because poliovirus translates a single polyprotein, we assume that all protein products are produced at equal rates based upon a single rate-constant of initiation. We also assume that poliovirus genomes, and not cellular factors, are rate-limiting, and neglect the delay between the initiation of translation and the appearance of the protein products. With these assumptions, translation can be modeled as a first-order equation with a single parameter, $c_{trans}$, yielding a rate of translation:

$$r_{trans} = c_{trans}\left[RNA^+_{lin}\right].$$ (3)

Here, and throughout the model, we consider these rates to describe Poisson processes, rather than changes in continuously valued quantities. Square brackets are used to indicate the concentration per cell, or abundance, of each molecular species. We track three protein products of translation: the procapsid units, which we abbreviate CAP, and protein products 3A and 3CD. Based on evidence from complementation experiments, we assume that CAP units and 3A diffuse freely, while 3CD accumulates within complexes with translating genomes (*Novak and Kirkegaard, 1994*; *Ansardi et al., 1996*; *Nugent et al., 1999*). *Equation 3* applies to translation of both complex-associated and free genomes. Global abundances of CAP and 3A are tracked, while abundances of 3CD are tracked individually for each replication complex. 3CD arising from the translation of free genomes is ignored.

## Replication complex formation (step 4)

We assume that two events must happen before a translating positive-sense strand can replicate: it must attach to a membrane, representing nucleation of a replication complex, and it must circularize through association with 3CD (*Gamarnik and Andino, 1998*; *Herold and Andino, 2001*). Once a strand associates with a membrane, we consider that it has formed a complex, and assume that all subsequent translation events will add to the local concentration of 3CD. We model this first step by introducing a rate, $r_{compart}$, at which the $RNA^+_{lin}$ species forms complexes. We also assume that the viral protein product 3A facilitates this complex formation (*Hsu et al., 2010*). Finally, we assume that complex formation is limited by the supply for suitable membrane, which limits the number of possible complexes in a cell to $com_{max}$ (*Guinea and Carrasco, 1990*). We therefore introduce a first-order reaction scaled by the number of existing complexes, com, the maximum, $com_{max}$, and the concentration of the protein 3A:

$$r_{com} = c_{com}\left(1 - \frac{[com]}{com_{max}}\right)[3A].$$ (4)

While other viral and cellular proteins are involved in complex formation, we assume that their influence is adequately represented by tracking the concentration of 3A. We also represent the

consumption of some number of 3A molecules in the formation of each complex by a parameter $c_{3A}$. If insufficient 3A is available upon complex formation, newly translated proteins are consumed by the existing complex until $c_{3A}$ have been allocated. Therefore, we are assuming that 3A binding is cooperative, and that incomplete complexes have much higher affinity for 3A than does the reaction to form a new complex.

## Circularization (step 5)

We model circularization—the transition of a positive-sense genome from a linear, translating molecule to a noncovalently associated circularized molecule competent for replication—as a first-order reaction driven by the concentration of the viral protein 3CD in each complex (indexed by $i$).

$$r_{circ}^i = c_{circ} \left[ 3CD \right]_i .$$ (5)

This formulation reflects experimental data supporting the direct role of 3CD in circularization (*Gamarnik and Andino, 1998*; *Herold and Andino, 2001*), and the low rate of rescue of 3CD-deficient strains by complementation in *trans* (*Novak and Kirkegaard, 1994*).

## Replication (step 6)

We distinguish replication rates for positive and negative strand synthesis with separate rate constants $c_{rep+}$ and $c_{rep-}$. We ignore polymerase concentrations and instead assume that both types of replication are first-order reactions modified by a common cellular resource limit. This limitation is parameterized by $rep_{max}$, the maximum number of replication events per cell permitted by some limited resource and implemented with a running counter, $rep$, of synthesized RNAs. We also assume that per-capita replication does not differ between replication complexes, allowing us to write a mass-action equation for both replication reactions.

$$r_{rep+} = c_{rep+} \left[ RNA_{circ}^+ \right] \left( 1 - \frac{rep}{rep_{max}} \right).$$ (6a)

$$r_{rep-} = c_{rep-} \left[ RNA_{circ}^- \right] \left( 1 - \frac{rep}{rep_{max}} \right).$$ (6b)

Note that $r_{rep+}$ measures the rate at which replication is initiated on positive-sense templates, producing negative-sense strands, and similarly, $r_{rep-}$ measures the rate of positive-strand production.

We allowed newly synthesized positive-sense genomes one of three fates: (1) associate with capsid protomers and become encapsidated, (2) diffuse into the cytoplasm where they can translate and potentially create new, independent compartments, or (3) remain in the complex in which they were generated and act as templates for further RNA replication. We assume that positive-sense genomes that remain in the complex are immediately competent for replication. We were unable to fit the sharp transition to exponential growth seen in our strand measurement data without allowing for this third option. Allowing newly synthesized positive-sense genomes to remain in the complex and act immediately as replication templates is consistent with previous reports indicating a coupling between translation and replication as we still require initial, infecting genomes to be translated before transitioning to replication (*Novak and Kirkegaard, 1994*). Negative-sense strands also stay in the complex in which they were produced and are immediately competent for replication.

We assume that only the positive-sense replication-competent form is packaged (*Ansardi et al., 1996*, but see; *Liu et al., 2010*), and that, following *Nugent et al. (1999)*, genomes can only be packaged as they are synthesized from a negative-sense strand. We therefore first determine whether the newly synthesized positive-sense strand is packaged; then, for unpackaged genomes, we calculate whether they remain in the replication complex.

## Packaging (step 7)

We assume that the rate of initiation of packaging is proportional to the global concentration of a virus-derived protein product, CAP, representing capsid protomers. These protomers form pentamers, of which 12 are required for each capsid; each packaging event therefore consumes 60 units of CAP. To account for the evidence that deficiencies in capsid proteins can be complemented in *trans*, we

allowed capsid proteins to diffuse freely throughout the cell. As with 3A molecules and complex formation, if a packaging event begins when the global abundance of CAP is less than 60, then further packaging is halted until this deficit is filled.

Using the approximation that each available CAP molecule independently contributes to the probability of packaging, we derive the probability that a newly synthesized positive-sense strand is packaged to be:

$$p_{pack} = 1 - e^{-c_{pack}[CAP]}. \tag{7}$$

## Positive strand dispersal (step 8)

The probability of a newly synthesized positive-sense strand to remain within its replication complex, assuming it was not packaged, is given by the parameter $c_{stay}$. The total probability is therefore:

$$p_{stay} = c_{stay}\left(1 - p_{pack}\right). \tag{8}$$

## Replication phenotype

Our primary goal is to infer the number of replication cycles between the infecting and the progeny virions. Defining a complete replication cycle to include both copying to a negative-sense strand, then back to a positive-sense strand, we label the mean of this value $\bar{g}$. The principal purpose of this mean is to link the mutation rate of replication to the mean mutation frequency in the progeny population, so the appropriate measure is to average over virions, not infected cells. For $k$ replicate simulations, let $n_i$ represent the number of progeny produced by each replicate $i$, and $g_{ij}$ represent the number of replication cycles in the ancestry of each virion $j$ in replicate $i$; then we calculate $\bar{g}$ as follows.

$$\bar{g} = \frac{\sum_{i=1}^{k}\sum_{j=1}^{n_i} g_{ij}}{\sum_{i=1}^{k} n_i}. \tag{9}$$

## **Parameter inference**

### Inference method

We chose to implement a version of Approximate Bayesian computation called Sequential Monte Carlo (*Sisson et al., 2007*; *Toni et al., 2009*; *Beaumont, 2010*; *Csilléry et al., 2010*; *Lopes and Beaumont, 2010*; *Toni and Stumpf, 2010*). This method consists of several rounds of parameter selection which form successively better approximations of the posterior distribution. In each round $x$, a population of size $n_x$ parameters sets is generated iteratively by choosing a parameter set from the preceding round $x − 1$, perturbing its values, then accepting or discarding the new parameter set based on the distance of its measured summary statistic from the summary statistic representing the data. Parameter sets with distances less than $\varepsilon_x$ are accepted; a diminishing series of thresholds, $\varepsilon_1 > \varepsilon_2 > \varepsilon_3$, etc, progressively focuses the search on those parameter values that best match the data. In round 1, parameter sets are drawn from the prior distributions; this first round is therefore identical to the basic rejection algorithm, but with a fairly large $\varepsilon_1$ to reduce computation time.

The advantage of searching for better parameter sets near previously identified good values is a much higher frequency of acceptance, and therefore much less computational time. However, the parameters accepted in later rounds are then biased toward common values in the previous rounds. The SMC algorithm removes this bias by weighting the selection of parameter sets against those that are most similar to their parent round, and toward those that resemble the prior distribution. Let $K_\sigma(\theta_a, \theta_b)$ represent the probability of perturbing $\theta_a$ into $\theta_b$ with a Gaussian kernel of standard deviation $\sigma$, $w_{xi}$ represent the weight of parameter set $i$ in round $x$, $\theta_{xij}$ represent the value of parameter $j$ in set $i$ of round $x$, and $\pi(\theta)$ represent the prior probability of $\theta$. Then, for round two and later, we calculate these importance weights as in *Equation 10* (adapted from *Toni et al., 2009*).

$$w_{x,i} = \frac{\sum_{j=1}^{10} \pi(\theta_{x,i,j})}{\sum_{k=1}^{n_{x-1}} w_{x-1,k} \sum_{j=1}^{10} K_{\sigma_{x-1,j}}(\theta_{x-1,k,j}, \theta_{x,i,j})}. \tag{10}$$

*Beaumont (2010)* suggests that the Gaussian perturbations applied to each proposed parameter set should be scaled with regard to the variance in that parameter in the previous round. In practice, we identified a trade-off based on the scaling of these perturbations; smaller perturbations lead to increased acceptance rates but more positively skewed importance weights; because the weighting of each accepted parameter set is normalized relative to the highest observed importance weight, this strong skew effectively dilutes the inferential power of the analysis. We found that using the standard deviation of each parameter as the standard deviation for the perturbation balanced this trade-off adequately for our model: the weights of the 1135 sampled parameter sets had an entropy of 10.01 bits, compared to a maximal value of 10.15 bits. This high entropy confirms that any remaining skew in the importance weights does not severely diminish the effective sample size.

Implementing this method requires several additional choices: the shapes of prior distributions, the number of rounds, the values of ε, and the number of replicates, *n*, to perform for each evaluation of the model. This last decision turned out to be crucial; inferring based on the mean of a larger number of replicates (n ≥ 1000) tended to select parameter sets with highly variable behavior. Reducing *n* led to a higher rate of parameter set rejection but more biologically plausible dynamics. A simple explanation for this pattern is that a number of parameter sets produce acceptable mean behavior, but differ in the degree of stochastic variation around that mean in a way that does not reflect measured stochastic variation (*Schulte and Andino, 2014*). We therefore chose to accept parameter sets that passed a given ε for multiple, sequential sets of *n* replicates. For round 1, one thousand parameters sets were accepted based on five sets of *n* = 20 replicates at MOI = 10 only with ε = 12. For round 2, 1135 parameters sets were accepted based on five sets of *n* = 20 replicates at MOI = 1 with ε = 16, five sets of *n* = 20 replicates at MOI = 10 with ε = 12, and five sets of *n* = 20 replicates at MOI = 100 with ε = 7. Thresholds for round 2 were calibrated to achieve an acceptance rate of about 1 in 10,000.

Approximate Bayesian computation uses summary statistics to judge the fit of model predictions to data. We chose to compute the sum squared error of the log of the mean abundances of positive- and negative-sense RNA, and the log ratio of positive-sense RNA in capsids to the total positive pool. This produced a single summary statistic that captured the total error of the predictions from a particular parameter set. Using the natural logs for each data point effectively weights each deviation by its relative magnitude, which prevents the large absolute size of errors at later time steps from dominating the error measurement.

To aid in visual exploration of the data, we chose five representative parameter sets as follows. From an initial batch of 513 parameter sets, we chose the 50 sets with the overall lowest error. From these, we sampled sets of five at random and calculated the summed pairwise distance in parameter space of those five sets from each other. To adjust for the different scales and uncertainties in each parameter, the contribution of each parameter to the distance measure was divided by its standard deviation over the whole set of 513 values. These summed distances provided a metric of the parameter diversity captured in a choice of five parameter sets; we examined one thousand randomly drawn sets of five and chose the set with the highest summed distance. These five parameter sets are shown in *Figure 3—source data 1*.

## Method validation

To test the effectiveness of the inference process and the adequacy of our data, we chose parameter sets from the prior, produced simulated data from these parameter sets, and then performed the sequential Monte Carlo method described above. We chose three sets of parameters based on two criteria: representation of a high diversity of the replication phenotype, $\bar{g}$, and biological plausibility. We achieved these criteria by sampling the prior, measuring $\bar{g}$ and the mean number of virions at MOI = 10, and choosing three parameter sets than spanned the range of $\bar{g}$ and produced a number of virions similar to the empirically measured value.

For each parameter set, we measured RNA abundances and, for MOI = 10, virion abundances with 10,000 replicate simulations. These data were then treated exactly as the empirical data were handled; the log of the averages was used to measure error in the Bayesian inference procedure described above. At least one thousand accepted parameter sets were collected for both inference rounds (except in one case in which fast convergence and lengthy computation times made round two unnecessary and computationally costly).

The mean RNA abundances for MOI = 10 and the inference results are plotted in *Figure 4—figure supplement 5*. In each case, the inference process produced a narrow posterior, relative to the prior,

with clear similarity to the actual value of $\overline{g}$ for each starting parameter set. However, the mean of the final posterior fails to perfectly match the true value in all three cases. Also, the second round of inference, which more than doubles the amount of data used to assess goodness-of-fit, achieved little in these validation experiments. The error thresholds (values of ε) used here may have been too permissive to achieve complete convergence to the correct value. Nonetheless, these experiments show that our statistical method, when combined with the types and quantities of data we have available, can produce a reliable inference.

## Sensitivity analysis

To further investigate if the parameter values identified by ABC minimize the error in predicted RNA and virion dynamics, we explored the sensitivity of mean error to variation in each parameter for a single parameter set ('Best' in *Figure 3—source data 1*). As before, error was assessed by comparing the log of the mean RNA and virion abundances to empirical data for sets of 20 simulations. 500 trials of 20 simulations each at each MOI were averaged to produce a high-resolution estimate of true deviation from the data. These results are plotted as $1/(1 + \text{mean error})$ to show an intuitive goodness-of-fit measure, where high values indicate similarity to the data. *Figure 4—figure supplement 4* shows that the 'best' parameter set is at or near a local maximum for goodness-of-fit for eight of ten parameters; the effects of the remaining two parameters, $c_{pack}$ and $com_{max}$, appear to be minimal for this parameter set. These results suggest that convergence of the posterior distributions is linked to the sensitivity of the model to each parameter, which supports the effectiveness of the ABC algorithm.

# Additional information

## Funding

| Funder | Grant reference number | Author |
| --- | --- | --- |
| National Institute of Allergy and Infectious Diseases | R01 AI36178 | Michael B Schulte, Raul Andino |
| National Institute of Allergy and Infectious Diseases | R01 AI40085 | Michael B Schulte, Raul Andino |
| Defense Advanced Research Projects Agency | BAA-10-93 | Michael B Schulte, Raul Andino |
| Burroughs Wellcome Fund | JBP | Joshua B Plotkin |
| David and Lucile Packard Foundation | JBP | Joshua B Plotkin |
| U.S. Department of the Interior | D12AP00025 | Joshua B Plotkin |
| Army Research Office | W911NF-12-1-0552 | Joshua B Plotkin |

The funders had no role in study design, data collection and interpretation, or the decision to submit the work for publication.

## Author contributions

MBS, JAD, Conception and design, Acquisition of data, Analysis and interpretation of data, Drafting or revising the article; JBP, RA, Conception and design, Analysis and interpretation of data, Drafting or revising the article

# Additional files

## Supplementary file

• Source code 1. Poliovirus replication mathematical model-code. Contains custom software used for simulations throughout this article.

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
