## [Decision Letter]

[Editors’ note: this article was originally rejected after review, but the authors were later invited to resubmit.]

Thank you for choosing to send your work entitled “Experimentally guided mathematical modeling reveals RNA virus replication principles shaping the mutation distribution” for consideration at *eLife*. Your full submission has been evaluated by Aviv Regev (Senior editor), a Reviewing editor, and 3 peer reviewers, and the decision was reached after discussions between the reviewers. We regret to inform you that your work will not be considered further for publication.

The three reviews are below for your consideration, with the following comments from the Reviewing editor.

This paper reports a new attempt to model viral replication, including stochastic effects from cell to cell, and pools of viral RNA that are in replicative vs. translation pools. The model fits data best with a geometric increase of RNA (no surprise) with about 5 generations occurring intracellularlly (before cell death?). The large number (ten or so) of parameters were explored by an Approximation Bayesian Computation method, not all were independent, and some having no major impact. While I cannot comment on the math, most of the assumptions of the model seem plausible and the predictions seem to fit the limited data. The model then is used to predict the distribution of mutations, with good correspondence to data.

While it seems like a useful framework has been devised, novel predictions of the model are few and far between.

I have a few issues that reflect my uncertainty about the working of the model.

1) I have a free-floating concern with identifying a cycle as a complete genome-to antigenome-to-genome cycle. How do we deal with the fact that there are many more genomes than antigenomes? In fact, this ratio is measured and predicted (Figure 3); wouldn't this be a good thing to include as a plotted output? Does one assume here that the excess of genomes are all moved out of the replication pool, while all the antigenomes are always in the pool? How do we deal with the necessary half-cycles? Some discussion of this would be helpful.

2) In the Results section, I might question the restriction that “following [32], genomes can only be packaged as they are synthesized from a negative-sense strand…”. Could we consider a more open view that any genome, new or old, could be packaged (from any pool)? Would this be compatible with the data? Was this tried?

Reviewer #1

Life history theory aims to quantify different parameters, such as time to reproduction or number of offsprings, in potential strategies in the life of organisms to maximize the number of surviving offspring. RNA viruses present a particularly simple lifestyle and their fast replication makes them particularly appropriate to test evolutionary models. The authors apply these principles to the evolution of poliovirus. This work is the collaboration between two strong groups in the area of mathematical modeling and experimental viral evolution. Of particularly interest in this work is the determination of parameters in single cell infection. For instance, the authors contrast two simple models of replication, stamping machine where the progeny is one generation away from the initial and the geometric replication model where several rounds of replication occur within a single cell. Main conclusions of this work the lifetime of viral progeny (∼5 generations) and the abundance RNA increase following an exponential pattern which allow to test different models of replication.

The mathematical model is a stochastic model a la Gillespie where the cell infection is decomposed into a set of individual steps (binding, uncoating, translation, replication complex formation, circularization, replication, packaging, strand dispersal). The model has many parameters. To infer the value of these parameters the authors use quantitative RT-PCR data of both positive-sense genomes and negative-sense, amounting to 27 measures. The stochastic approach coupled to high dimension of data and large numbers of parameters led the authors to choose an approximate Bayesian computation using as summary statistic the sum of the squared deviations of the average simulated RNA concentration.

Initially I was extremely excited about the manuscript, but I became less enthusiastic as I was reading along.

1) The model proposed discretized the infection into a set of simple steps each of this depending on a few parameters, creating a complex chain of steps. Each step has its own parameters that add complexity in addition to the challenges of stochasticity. The model fits many parameters. It would be more interesting to make some specific hypotheses to be tested experimentally beyond parameter fitting.

2) The steps follow a virology book cartoon model, but many other factors are known to be important (transport, cell factors, etc.), and it is unclear that the factors considered are the only important ones.

3) The variability among cells could be large, as reflected by recent single cell studies, and it is unclear how this variability is taken into account in these models. A critical question in this work is the cell to cell variability. In a system where there is geometric growth a progeny the mean number of viruses in a cell is not as informative as the whole distribution. One would think that the larger scale dynamics of the virus are determined by the cells producing more viruses, in particular, in a geometric growth model.

Reviewer #2

The authors have built on their previous work in poliovirus cell interaction to develop an elaborate model of intracellular replication and consequent accumulation of mutations. In combination with careful experimental analysis of replication kinetics of both plus and minus strands as well as virions at several different MOIs, they are able to establish reasonably tight ranges for the 10 or so free parameters required by their model and draw several interesting conclusions, including that intracellular replication is best fit to a branching (rather than “stamping machine”) process, that, on average, progeny of an infected cell are about 5 copying generations removed from the infecting genome, and that, therefore, the underlying mutation rate (per copying event) is about 5-fold less than estimated from single-step experiments.

Although the modeling analysis seems to me to be quite elegant, to use up-to date techniques, and to be well supported by careful experimentation, and the conclusions seem reasonable and well justified, I am not an expert in either modeling or positive-strand virus replication, so I will defer to other reviewers on these issues. That said, my major concern is that the manuscript is rather inaccessible to the general readership expected for a journal like *eLife*. I found it quite difficult to read and understand, and I had to go back to a virology textbook a few times. It would help a lot if Figure 2 contained more information for the general reader, such as a map of the genome indicating the names and functions of the important gene products. Also very helpful would be a summary table listing the various parameters and the steps they refer to. Overall, the reader needs to have a clearer vision of the replication cycle than is accessible from the introductory material here. As another example, the description of the necessity for “circularization” of the RNA, and the drawing in Figure 2, leave the non-expert reader with the strong impression that the circles referred to are covalently bound RNA-only structures, rather than RNA-protein complexes, as the authors have nicely shown in previous work.

Although the stated goal of the analysis is to understand how the replication mode shapes the mutation distribution, the treatment of mutations seems rather lightweight, by comparison to replication dynamics, and also has confusing aspects. For starters, in most modeling I am accustomed to, μ is used to refer to the mutation rate, and s the selection coefficient. The use of different notation here made reading more difficult, for sure. Second, the use of a “one size fits all” approach to mutations seems oversimplified and inaccurate, because the kinetics of accumulation of different types of mutants will depend very strongly on how and where they act, whether in cis or trans to the genome, whether they affect one protein (e.g. missense mutations) or others (nonsense mutations), etc. If and how the authors included these issues in their simulations is not made clear. Obviously, there must be accumulation of some mutations that would be lethal in clonal replication, but not others, so the meaning of “replication deficit” is quite unclear.

Reviewer #3

Schulte et al. present a detailed analysis of the replication process of polio virus inside a cell. They develop a stochastic model of viral replication, fit it to empirical data via Approximate Bayesian Computation (ABC), and infer several parameters of interest. Most importantly, they show that polio virus does not replicate using the stamping-machine mechanism. Instead, there are intermediate rounds of replication inside the cell.

Overall, the paper is interesting, and the authors have a wealth of novel results. However, the presentation of the work is lacking at places, and the paper will require substantial revisions before it is publishable.

Most importantly, I feel the authors can't entirely decide on what their story is. Is the main motivation to identify the replication mode of the virus, or to show that a stochastic model is better than a deterministic model? At places, the paper seems to argue strongly for stochastic over deterministic treatment, but in the end there isn't a single result I could see that would demonstrate how a deterministic approach fails. I think it would be better to focus on the biological question (method of replication mode) and not worry so much about justifying the stochastic approach.

Secondly, the computational methods are not that well described. Notably, the methods section talks only about experimental work. Yet the main body of the text is not sufficient to fully understand the computational work, and some paragraphs currently in the Results would be better placed in the Materials and methods (see below). Also, the code and raw data for this project need to be made available.

Other comments:

I couldn't always figure out what parts of the work were simulation and what parts were fitting of the model to measured data. This needs to be worked out more clearly. Also, it would be good to have a table that lists the final parameter estimates obtained from fitting the model to the data.

It would be good to add a step of model verification where you simulate the model given some parameter set, and then see whether you can recover the parameters using the ABC method. In particular, can you distinguish SM and GR scenarios when all other parameters are the same?

In the Introduction: I would argue that if a deterministic model agrees well with measured data then that is sufficient evidence to conclude the stochastic fluctuations can be neglected.

In the Results: The modified Gillespie algorithm needs to be explained in detail in the Materials and methods section. For example, what is the threshold parameter at which you switch from stochastic to deterministic treatment? Also, the code needs to be made available.

In the Results, “we also assume that poliovirus genomes, and not cellular factors, are rate-limiting”: it is not clear to me to what extent this assumption biases results. Certainly, eventually cellular factors will be limiting viral replication inside a cell. So how does your model take this into account?

In the Results, “inferring based on mean of a larger number of replicates (n ≥ 1000) tended to select parameter sets with highly variable behavior. Reducing n led to a higher rate of parameter set rejection but more biologically plausible dynamics”: any idea why your results depend on n in this way?

Figure 1: Where does the number of 2.33 come from for GR? I doubt it is part of the definition for GR, but the sentence is written as if it were.

---

## [Author Response]

We have incorporated many of the reviewers’ suggestions and modified the paper to clarify several aspects that were unclear in the previous version. We have also included additional information and novel analysis of our data. We believe that the revised version is much stronger now.

Critical and innovative advances in our study are:

1) This is the first stochastic model for any virus replication that uses experimental information to infer the replication parameters;

2) We clearly establish that poliovirus replication significantly deviates from the accepted “stamping machine” strategy;

3) We use this model to examine how the genetic structure of a virus population is established during replication in single infected cells.

*1) I have a free-floating concern with identifying a cycle as a complete genome-to antigenome-to-genome cycle. How do we deal with the fact that there are many more genomes than antigenomes? In fact, this ratio is measured and predicted (*Figure 3*); wouldn't this be a good thing to include as a plotted output? Does one assume here that the excess of genomes are all moved out of the replication pool, while all the antigenomes are always in the pool? How do we deal with the necessary half-cycles? Some discussion of this would be helpful*.

We have revised our manuscript to use the word “generation” in place of “cycle”—this change in terminology should help clear up this issue for most readers. We use “generation” to denote the completion of a full round of the intracellular replication process of a genome produced from a negative-sense strand that itself arose from a positive-sense genome. This definition does not assume that genomic and anti-genomic molecules must balance or have any particular stoichiometry. Of course, the premise of our model is that the relative abundances over time of genomic and antigenomic RNAs are informative about replication, and those abundances are the primary data we fit (Figure 3). However, each genomic RNA is related to the progenitor genomes by some number of generations, each consisting of a genomic-to-antigenomic step, then a corresponding antigenomic-to-genomic step. It is the counts of these generations, not the ratios, which are plotted in Figure 3, and which define the opportunity for mutation. There is therefore no contradiction among our results: the balance of rates of replication and other processes determine the ratios, and also determine the mean number of generations, which we now call g¯.

*2) In the Results section, I might question the restriction that “following*
[32]*, genomes can only be packaged as they are synthesized from a negative-sense strand…”. Could we consider a more open view that any genome, new or old, could be packaged (from any pool)? Would this be compatible with the data? Was this tried*?

Rather than try every possible derivation of our model, we choose to build off the wealth of insights to poliovirus biology found in the literature. In addition to [32] showing a coupling of replication and packaging, [3] show an increase in viral particles only in the presence of active replication, suggesting that genomes destined for viral particles are siphoned off of the replication machinery.

Reviewer #1

*1) The model proposed discretized the infection into a set of simple steps each of this depending on a few parameters, creating a complex chain of steps. Each step has its own parameters that add complexity in addition to the challenges of stochasticity. The model fits many parameters. It would be more interesting to make some specific hypotheses to be tested experimentally beyond parameter fitting*.

Our primary result—that around five generations separate a typical progeny from the infecting genome—is a specific and testable hypothesis, and we have sought to emphasize that point more explicitly. To clarify, this result provides a quantitative prediction of the ratio of the frequency of a neutral mutation after a generation to the rate of such mutations. We are working to develop methods to test these predictions and hope that the publication of these results will stimulate other such experiments. We also produce these predictions for a range of deleterious mutations (Figure 5). We have revised the presentation of these results to make this point more clearly.

*2) The steps follow a virology book cartoon model, but many other factors are known to be important (transport, cell factors, etc.), and it is unclear that the factors considered are the only important ones*.

We agree completely with this conclusion, which is why it is critical to provide testable predictions that could point to inadequacies in our model. As noted above, the model is already complex; we believe that the best approach is to construct a model with minimal complexity that both captures the essential biological steps and recapitulates the measured dynamics (as seen in Figure 3). Publishing this model will provide a much-needed testbed for researchers to assess the quantitative contribution of the many other details of poliovirus replication.

*3) The variability among cells could be large, as reflected by recent single cell studies, and it is unclear how this variability is taken into account in these models. A critical question in this work is the cell to cell variability. In a system where there is geometric growth a progeny the mean number of viruses in a cell is not as informative as the whole distribution. One would think that the larger scale dynamics of the virus are determined by the cells producing more viruses, in particular, in a geometric growth model*.

We agree that stochastic variability is extremely important—our revised Materials and methods section hopefully do a better job of communicating the details behind our stochastic model.

Reviewer #2

*Although the modeling analysis seems to me to be quite elegant, to use up-to date techniques, and to be well supported by careful experimentation, and the conclusions seem reasonable and well justified, I am not an expert in either modeling or positive-strand virus replication, so I will defer to other reviewers on these issues. That said, my major concern is that the manuscript is rather inaccessible to the general readership expected for a journal like eLife. I found it quite difficult to read and understand, and I had to go back to a virology textbook a few times. It would help a lot if*
Figure 2
*contained more information for the general reader, such as a map of the genome indicating the names and functions of the important gene products. Also very helpful would be a summary table listing the various parameters and the steps they refer to. Overall, the reader needs to have a clearer vision of the replication cycle than is accessible from the introductory material here. As another example, the description of the necessity for “circularization” of the RNA, and the drawing in*
Figure 2*, leave the non-expert reader with the strong impression that the circles referred to are covalently bound RNA-only structures, rather than RNA-protein complexes, as the authors have nicely shown in previous work*.

Thank you for these positive comments. We have extensively revised the Materials and methods section and believe it now gives a more complete picture of both the model and the life-cycle. We have revised our description of the “circular” RNA form to state its noncovalent nature explicitly.

*Although the stated goal of the analysis is to understand how the replication mode shapes the mutation distribution, the treatment of mutations seems rather lightweight, by comparison to replication dynamics, and also has confusing aspects. For starters, in most modeling I am accustomed to, μ is used to refer to the mutation rate, and s the selection coefficient. The use of different notation here made reading more difficult, for sure. Second, the use of a “one size fits all” approach to mutations seems oversimplified and inaccurate, because the kinetics of accumulation of different types of mutants will depend very strongly on how and where they act, whether in cis or trans to the genome, whether they affect one protein (e.g. missense mutations) or others (nonsense mutations), etc. If and how the authors included these issues in their simulations is not made clear. Obviously, there must be accumulation of some mutations that would be lethal in clonal replication, but not others, so the meaning of “replication deficit” is quite unclear*.

We have changed the notation for the number of generations to avoid the ambiguity of *μ*. We absolutely agree that different types of mutations would accumulate differently, and have expanded the text to describe this issue and justify our modeling of a particular class of mutation—those that directly affect intracellular replication rates.

Reviewer #3

*Most importantly, I feel the authors can't entirely decide on what their story is. Is the main motivation to identify the replication mode of the virus, or to show that a stochastic model is better than a deterministic model? At places, the paper seems to argue strongly for stochastic over deterministic treatment, but in the end there isn't a single result I could see that would demonstrate how a deterministic approach fails. I think it would be better to focus on the biological question (method of replication mode) and not worry so much about justifying the stochastic approach*.

You are correct—the point of the paper is not to argue for the superiority of a stochastic model, although we think it is both useful and necessary to differentiate our work from previous models. We have edited the Introduction and Discussion to shift the emphasis to our main result, namely the prediction that poliovirus follows the geometric replication mode, i.e. multiple iterative generations of genomic replication produce progeny virus.

*Secondly, the computational methods are not that well described. Notably, the methods section talks only about experimental work. Yet the main body of the text is not sufficient to fully understand the computational work, and some paragraphs currently in the Results would be better placed in the Materials and methods (see below). Also, the code and raw data for this project need to be made available*.

*Other comments*:

*I couldn't always figure out what parts of the work were simulation and what parts were fitting of the model to measured data. This needs to be worked out more clearly. Also, it would be good to have a table that lists the final parameter estimates obtained from fitting the model to the data*.

We have reorganized the Materials and methods and the Results to help clarify these and other issues. We have also reorganized the figure supplements to make this kind of information easier to find. The full posterior distribution for each parameter is graphed in Figure 4—figure supplement 1, and point estimates are given in Figure 3—figure supplement 2.

*It would be good to add a step of model verification where you simulate the model given some parameter set, and then see whether you can recover the parameters using the ABC method. In particular, can you distinguish SM and GR scenarios when all other parameters are the same*?

This is an excellent suggestion and we have added a figure doing exactly this (Figure 5—figure supplement 5).

*In the Introduction: I would argue that if a deterministic model agrees well with measured data then that is sufficient evidence to conclude the stochastic fluctuations can be neglected*.

Our view is that the biological process is clearly stochastic, making a stochastic model a reasonable starting place. However, we agree that the original statement was misleading—if a deterministic data was found to adequately explain all data, then a stochastic model would be unnecessary.

*In the Results: the modified Gillespie algorithm needs to be explained in detail in the Materials and methods section. For example, what is the threshold parameter at which you switch from stochastic to deterministic treatment? Also, the code needs to be made available*.

This parameter value, as well as other missing details, have been added to the thoroughly revised Materials and methods section. The code has also been made available.

*In the Results, “we also assume that poliovirus genomes, and not cellular factors, are rate-limiting”: it is not clear to me to what extent this assumption biases results. Certainly, eventually cellular factors will be limiting viral replication inside a cell. So how does your model take this into account*?

We do implement a limitation on the replication of poliovirus genomes as the parameter *rep*_max_, and an additional limitation on the number of replication complexes as the parameter *com*_max_. We found that the addition of these parameters allowed the replication rate to decline toward the end of an infection, in keeping with the empirical data (and common sense). Without evidence that resources specifically limited translation, we didn’t see the need to add this complicating parameter to the model.

*In the Results, “inferring based on mean of a larger number of replicates (n ≥ 1000) tended to select parameter sets with highly variable behavior. Reducing n led to a higher rate of parameter set rejection but more biologically plausible dynamics”: any idea why your results depend on n in this way*?

We have added the text: “A simple explanation for this pattern is that a number of parameter sets produce acceptable mean behavior, but differ in the degree of stochastic variation around that mean in a way that does not reflect measured stochastic variation (42).”

Figure 1*: Where does the number of 2.33 come from for GR? I doubt it is part of the definition for GR, but the sentence is written as if it were*.

The number refers to the particular example plotted on the right side of Figure 1. The misleading sentence has been rephrased to indicate this.